# Specific and Generic Immunorecognition of Glycopeptide Antibiotics Promoted by Unique and Multiple Orientations of Hapten

**DOI:** 10.3390/bios9020052

**Published:** 2019-04-04

**Authors:** Maksim A. Burkin, Inna A. Galvidis, Sergei A. Eremin

**Affiliations:** 1Immunology Department, I. Mechnikov Research Institute for Vaccines and Sera, 105064 Moscow, Russia; galvidis@yandex.ru; 2Faculty of Chemistry, M. V. Lomonosov MSU, Leninsky Gory, 1, 119991 Moscow, Russia; eremin_sergei@hotmail.com

**Keywords:** hapten conjugate design, antibody repertoire, group recognition, ristomycin, teicoplanin, eremomycin, vancomycin

## Abstract

Conjugation chemistry does not always provide adequate spatial orientation of hapten in immunogens for the best presentation of generic or individual epitopes. In the present study, the influence of unique and multiple orientations of immunizing hapten on the immune response repertoire was compared to select generic recognition system. The glycopeptides, teicoplanin (TPL) and ristomycin (RSM), were conjugated to BSA to produce immunogens with unique and multiple orientations of haptens. Polyclonal antibodies generated against TPL conjugated through a single site were of uniform specificity and demonstrated selective TPL recognition, regardless of the coating conjugates design. The sensitivity (IC_50_) of 4 enzyme-linked immunosorbent assays (ELISAs) for TPL varied little within the 3.5–7.4 ng/mL, with a dynamic range of 0.2–100 ng/mL. RSM was coupled to BSA through several glycoside sites that evoked a wider repertoire of response. This first described anti-RSM antibody was selective for RSM in homologous hapten-coated ELISAs with IC_50_ values in the range 4.2–35 ng/mL. Among the heterologous antigens, periodate-oxidized TPL conjugated to gelatine was selected as the best binder of generic anti-RSM fraction. The developed ELISA showed group recognition of glycopeptides RSM, TPL, eremomycin, and vancomycin with cross-reactivity of 37–100% and a 10–10,000 ng/mL dynamic range. Thus, multiple presentations of immunizing hapten help expand the repertoire of immune responses and opportunities for the selection of the required fine-specificity agent.

## 1. Introduction

Currently, immunoanalytical techniques are wide-spread in various spheres of investigations. The majority of these methods are inexpensive, easy to handle, sensitive enough to determine trace amounts of the analyte, and suitable for a high-throughput analysis of a large number of samples. Because of these advantages, these methods are used in food safety and authentication analysis [1,2], the monitoring of environmental contamination and diseases markers [3,4], forensic examinations, and exploratory research screening [5,6]. The significant variety and abundance of natural and synthetic substances, used as target analytes, encourage the development of multidetection methods. A very useful feature for increasing the efficacy of screening methods is the capability to recognize several structural analogues, their derivatives and active metabolites [7,8,9,10]. Possessing a broad specificity, such methods allow for the detection of any representative of the group of related compounds in the test sample. There are several known approaches that allow for the development of antibodies specific to common molecular determinants and the achievement of broad immunodetection:Targeting an immune response against common molecular determinants using a rationally designed immunizing hapten [8,11], keeping in mind the naturally produced antibodies;Screening combinatorial immune or synthetic libraries to target generic epitopes and improve the properties of recombinant antibodies, using maturation methods [12,13];Combining several antibodies with specificity that complement each other in a hybrid assay [14] or designing a bispecific antibody [15];Additional facilities for group recognition of analytes by polyclonal antibodies can be provided by the affinity fractionation approach, using the principle of hapten heterology. A structurally related analyte as a heterologous immobilized hapten can bind only to the portion of antiserum antibodies directed against common epitopes. The assay developed according to the mentioned approach demonstrates improved group specificity, in comparison with homologous assay formats [16,17,18].

For example, the affinity fractionation was applied for polyclonal antibodies that initially showed selectivity for the individual glycopeptide antibiotic, eremomycin (ERM) [16]. However, the structural analogue, vancomycin (VCM), presented on the coating antigen, made it possible to bind a part of the antibodies to the common moieties of these glycopeptides. This within-assay selection of the group-specific antibody fraction provided the capability for group determination of both analytes. The other related glycopeptides, such as teicoplanin (TPL) and ristomycin (RSM), were not recognized in the reported enzyme-linked immunosorbent assay (ELISA).

Glycopeptide antibiotics begin their history in the mid-1950’s, when VCM and RSM were discovered. Now, the family of glycopeptide antibiotics includes a number of clinically approved representatives and a number of derivatives under research. VCM; TPL; and, more recently, telavancin, dalbavancin, and oritavancin have been approved as therapeutics for the treatment of gram-positive infections [19].

ERM is the nearest and more active analogue of VCM, isolated from a culture of *Nocardia orientalis* [20]. ERM has an additional glycoside substituent but lacks one chlorine atom, unlike VCM (Figure 1). TPL and RSM form another subgroup of glycopeptides, as their structures are more glycosylated and include a lipid substituent, in comparison with antibiotics from the VCM-group. RSM (also known as ristocetin) is produced by *Amycolatopsis lurida* [21]. At present, RSM has limited usage as a therapeutic agent because of its platelet agglutination activity. However, because of this feature, RSM has found usage in the diagnosis of hereditary haemorrhagic thrombocytopathies, von Willebrand disease, and Bernard–Soulier syndrome [22].

Despite the individual structural features, the drugs in both subgroups inhibit the growth of gram-positive bacteria and have the same mechanism of action. This class of drugs inhibits the synthesis of cell walls in sensitive microorganisms. Binding to the D-alanyl-D-alanine ends (D-Ala-D-Ala) of growing peptidoglycans, glycopeptides prevent the polymerization of peptidoglycan and disrupt the synthesis of the cell wall of gram-positive bacteria [23,24]. Thus, the common mode of glycopeptide reception suggests the presence of similar functional structures, which can serve as a target for generic binders. Among the antibody-based methods; polarization fluorescent immunoassay [25,26,27]; radioimmunoassay [25]; immunofluorescence [27,28]; and ELISA [29] for VCM, TPL, and ERM have been developed. The principle of specific interaction with D-Ala-D-Ala was the basis for the sandwich immunoassay of TPL [30].

To the best of our knowledge, there have been no reports concerning antibodies to the other glycopeptides or immunoassays performing group specificity. So, in the present study we attempt to use RSM as a novel hapten and TPL for the generation of antibodies and the development of a group-specific assay for glycopeptide antibiotics. The effect of immunizing hapten unique- or multi-presented on the immunogen has been observed in this work. The expanded repertoire of immune response, promoted by the multi-presentation of the immunizing hapten, contributed to a wider choice of suitable candidates for the generic reagent. Coating conjugates developed on the basis of heterologous haptens were applied here as affine binders for the selection of antibodies against epitopes common to the main analyte and its analogue. The mentioned approach was exactly the right tool that allowed the generic immunorecognition tuning.

## 2. Materials and Methods

### 2.1. Chemicals

Ristomycin A (RSM), teicoplanin A2 (TPL), eremomycin (ERM), and vancomycin (VCM) were kindly provided by the Gause Institute of New Antibiotics.

Bovine serum albumin (BSA), gelatine (Gel), N-hydroxysuccinimide (NHS), 1-ethyl-3-(3-dimethylaminopropyl)carbodiimide (edc), sodium periodate (pi), formaldehyde (f), glutaraldehyde (ga), and Freund’s complete adjuvant were purchased from Chimmed (Moscow, Russia).

### 2.2. Synthesis of Conjugated Antigens


*Gel-RSM(ga), Gel-TPL(ga)*


Two solutions of Gel (3 mg, 19 nmol) in 0.5 mL of distilled water were mixed with 25-fold molar excesses of RSM [2068] or TPL [1879] (0.98 mg and 0.89 mg, respectively). Freshly prepared 2.5% solution of glutaraldehyde (ga), 30 µL, was added to each mixture and stirred for 2 h at room temperature. Then, 0.1 mL of sodium borohydride (2 mg/mL) was supplemented and stirred for another 2 h. The resultant conjugates were dialyzed to remove the excess reagents.


*Gel(pi)-RSM, Gel(pi)-TPL*


Crystalline sodium periodate (2.4 mg, 114 nmol) was added to Gel solution (18 mg, 2 × 57 nmol) in 2 mL of 10mM acetic buffer (pH 5.0) and stirred for 15 min using a magnet stirrer. Oxidized glycoprotein was dialyzed against 10mM acetic buffer (pH 5.0) overnight at 4 °C. The resultant volume was divided into two portions, which were dropwise added to 2 mL-solutions of RSM (2.95 mg) and TPL (2.68 mg) in a 0.1 M carbonate-bicarbonate buffer (CBB, pH 9.6). Mixtures of proteins and haptens prepared using a molar ratio of 1/25 were stirred for 2 h and for an additional 2 h after addition of sodium borohydride (0.1 mL, 2 mg/mL). The resultant conjugates were dialyzed exhaustively against phosphate buffered saline (PBS, pH 7.2).


*Gel-RSM(edc), Gel-TPL(edc)*


Two solutions—RSM (1.3 mg, 0.625 µmol) and TPL (1.2 mg, 0.625 µmol), each in 1 mL of distilled water—were supplemented with 15 mg EDC and stirred for 30 min. Then, solutions of activated RSM and TPL were added dropwise to Gel (4 mg, 25 nmol) in 0.5 mL of CBB and stirred overnight. Unreacted reagents were removed from the resultant conjugates using dialysis.

*Gel-RSM(ae), Gel-TPL(ae),* and *BSA-TPL(ae)*

The mixtures of RSM (6.2 mg, 3 µmol) or TPL (5.64 mg, 3 µmol) with NHS (0.52 mg, 4.5 µmol) and EDC (0.87 mg, 4.5 µmol) in 1 mL dimethylformamide were composed and stirred 1.5 h at room temperature, using a magnet stirrer. Then, the active ester (ae) glycopeptide intermediates were dropwise added to Gel (3 mg, 19 nmol) or BSA (4 mg, 60 nmol) in CBB. The mixtures, prepared using the molar ratios 1/25 and 1/50 for Gel/hapten and BSA/hapten, respectively, were stirred overnight and then dialyzed.


*Gel-RSM(f), Gel-TPL(f)*


Protein solutions containing Gel (3 mg, 19 nmol) in 0.5 mL of water were combined with water solutions of RSM or TPL (0.98 mg and 0.89 mg, 475 nmol, respectively). Each protein–hapten mixture was supplemented with 0.3 mL of 37% formaldehyde (3.69 mmol) and stirred overnight at 37 °C, followed by exhaustive dialysis.

*Gel-RSM(pi), BSA-RSM(pi),* and *Gel-TPL(pi)*

Equimolar and 3-fold molar quantities of sodium periodate (103 and 309 µg from 1 mg/mL solution) were added to 0.5 mL water solutions containing 475 nmol of RSM (2 × 982 µg) or TPL (2 × 688 µg) and stirred for 20 min. Oxidized glycopeptides were added to Gel (3 mg, 19 nmol) in 0.5 mL CBB and stirred for 2 h. Then, 0.1 mL of sodium borohydride (2 mg/mL) was added to each reaction mixture, and 2 h later, the low-molecular-weight components of each reaction mixture were removed by dialysis. The Gel/antibiotic molar ratio was taken as 1/25. The same procedure was conducted for BSA conjugates, for which the molar ratio of components consisted 1/50.

All of the products of conjugation were transferred into dialysis tubes with molecular weight cut-off of 14,000 Da and dialyzed against 3 exchanges of 5 L PBS (pH 7.2) for 2 days at 10 °C. The prepared conjugates were stored at −15 °C as 1 mg/mL-solutions in 50% glycerol. Conjugate formation was confirmed by spectral characteristics obtained using a spectrophotometer Shimadzu 1800 (Shimadzu, Japan).

### 2.3. Immunization and Polyclonal Antibody Preparation

Chinchilla rabbits (2.0–2.5 kg) were kept and treated in accordance with the guidelines for the care and use of laboratory animals [31]. The experimental protocol of the present study was fully approved by the Bioethics Committee for Animal Care of I. Mechnikov Research Institute for Vaccines and Sera.

The schedules of immunization with BSA-RSM(pi-3) and BSA-TPL(ae) were similar. Rabbits were injected subcutaneously with 100 µg of conjugates emulsified in complete Freund’s adjuvant. Repeated administration of the immunogens was performed monthly with the same dose of conjugates in saline. A week after each repeated injection, a sample of blood was taken from the marginal ear vein, the serum was separated, and an equal amount of glycerol was added, to store the antibody at –15 °C until use.

### 2.4. ELISA Procedure

Indirect antigen-coated enzyme-linked immunoassay was executed according to the procedure described before [32]. Optimal reagent concentrations were determined using check board titration. The coating antigen–antibody pairs was taken in concentrations that provided absorbance of about 1.0 and examined in a competitive assay.

All the prepared Gel-glycopeptide conjugates were coated on high-binding polystyrene Costar 96-well plates in 0.1 mL of CBB (pH 9.5) and incubated at 4 °C overnight. After the plates were washed 3–5 times with PBS containing 0.05% Tween 20 (PBS-t), 0.1 mL of analyte standard (10,000–0.01, and 0 ng/mL) was added. Then, 0.1 mL of antibody in PBS-t with 1% BSA was added to each well and the plates were incubated at 25 °C for 1 h. After washing, the bound antibodies were detected using the secondary anti-rabbit IgG antibodies conjugated with horseradish peroxidase (1 h, 37 °C). The enzymatic reaction was initiated by adding 0.1 mL of TMB substrate (Bioservice, Moscow, Russia) and terminated 30 min later by adding 0.1 mL of 1 M sulphuric acid. The absorbance was read at 450 nm using a StatFax 2100 (Awareness Technologies, Westport, CT, USA). The maximal antibody binding level occurring at a zero analyte concentration (B_0_) was taken as 100%. For each antibiotic concentration, the ratio B/B_0_ × 100 was calculated and used for plotting the standard curves. The half-inhibition concentration of analyte (IC_50_) served as the sensitivity value for comparison of assay variants. The cross-reaction of glycopeptide analogues was determined as a percentage of IC_50_ values for the main analyte and structural analogue. The limit of analyte detection (LOD) was accepted as 20% of the inhibitory concentration value (IC_20_), and the dynamic range of assay was IC_20_–IC_80_ [33].

## 3. Results and Discussion

### 3.1. Synthesis and Characterization of Conjugated Antigens

The molecules of glycopeptide antibiotics exposed a number of functional groups that allowed conjugation to protein, using different sites, and provided various orientations on the carrier (Figure 1).

The amino groups RSM and TPL were coupled with the amino groups of proteins, using glutaraldehyde, or with periodate-oxidized hydroxyls on glycoprotein, as a result of the reductive amination process. In the first case, the molecules of hapten and its carrier were shared with the C5-spacer; the second procedure provided the zero-length spacer [34]. Due to several available amino groups in RSM and TPL molecules, the structure of the resulting conjugates could be heterogeneous.

For conjugation via the carboxyl group of haptens, carbodiimide condensation and reaction of active esters were used. Both reactions are known to provide formation of the same stable amide bond, but the latter is faster and more effective [34]. The only carboxyl in TPL involved in the coupling provided a site-specific conjugation and a strictly defined hapten orientation on the carrier.

In the case of RSM, the carboxyl group is replaced by carboxymethyl, which cannot be activated by EDC. Although, EDC reactivity shows a strong preference towards the carboxyl group, in its absence, the possible target of carbodiimide activation may be a phenolic group, a potential site for coupling [35]. Phenolic and resorcylic groups of RSM and TPL could also interact with protein amines under the action of formaldehyde [29]. However, these sites are not specific, and the Mannich reaction may involve the other hydrogen active groups [34].

The carbohydrate component of RSM and TPL under the action of NaIO_4_ was oxidized, and the resultant aldehydes reacted with amines of protein molecules. According to our previous experience with the carbohydrate-containing hapten, lincomycin [36], the ratios between hapten and sodium periodate were taken as 1/1 and 1/3, to prevent excessive degradation of the sugars.

The formation of the prepared conjugates was confirmed by their UV spectra and also by immunizing capacity and immunochemical activity. The spectra of immunogens BSA-RSM (pi3) and BSA-TPL (ae) (Figure 2) showed an augmentation of typical peaks of RSM (λ = 280 nm) and TPL (λ = 278 nm) in comparison with BSA.

Six applied coupling procedures provided conjugates with five various chemical bonds between the haptens and the carriers (edc = ae) and at least four attachment sites in hapten molecules corresponding to the reactive groups, namely, amines, carboxyl, phenolic/resorcylic, and oxidized glycoside fragments (Table 1). The resultant various spatial arrangements of glycopeptide in the coating conjugates led to the presentation of different hapten epitopes on the antigens. The latter were able to bind with different portions of antiserum antibodies, which caused the changes in cross-reactivity demonstrated below.

### 3.2. Preparation of Polyclonal Antibodies

In choosing the immunogen design, we pursued two goals: (1) to assess the specificity of the immune response when the common or individual determinants of hapten were exposed on immunogen and (2) to investigate the effect of strictly defined and multiple presentations of hapten on the multi-epitope specificity of generated antibodies and on the ability to control the specificity of immunoassay.

Despite the general structural similarity between RSM and TPL, some areas of the molecules have individual features (Figure 1), especially those expressed in the upper part of the formulas: a long aliphatic substituent in TPL and a large tetrasaccharide fragment in RSM. To mask these distinctive molecule fragments by a carrier and expose common structural moiety of haptens, a conjugation involving the tetrasaccharide component of RSM was undertaken. In addition, an oxidation of carbohydrates using a more destructive 3-fold excess of sodium periodate (pi-3 vs pi-1) was used for preparation of the BSA-RSM(pi-3) immunogen.

In contrast, generation of anti-TPL antibodies was supposed, to exhibit individual determinants of hapten on the immunogen. To meet this requirement, a carboxyl group in TPL was chosen as a convenient site of conjugation, and the method of activated esters was used to prepare BSA-TPL(ae) immunogen.

Both conjugates that were intended for immunization, BSA-RSM(pi-3) and BSA-TPL(ae), served as examples of multiple orientation and strictly defined presentation of hapten on the carrier, respectively. Several opposite sites on the RSM carbohydrates (Figure 1, marked with arrows) could be oxidized after sodium periodate treatment, allowing for the possibility of differently oriented hapten coupling. In contrast, a single carboxyl in TPL and a reliable coupling method provided a specific conjugation (Figure 1, marked with arrow).

Maturation of the immune response and changes in the properties of antibodies were monitored during the immunization course and were evaluated in competitive ELISAs on the panel of homologous hapten-based antigens. The overall trend of improving sensitivity parameters during the course of immunization was noted. According to these observations, the highest sensitivity was achieved for both antibodies after the fifth immunization. The sensitivity values of corresponding ELISA variants are indicated in Figure 3 and Figure 4. As a result, the following experiments on the specificity of anti-BSA-RSM(pi-3) and anti-BSA-TPL(ae) were conducted with selected antisera.

### 3.3. Examination of Assay Specificity Based on Prepared Immunoreagents

An attempt to select suitable candidates for the generic reagent from the antibody repertoire of obtained antisera was conducted using different designs of the coating antigen. Different hapten epitopes on the carrier caught different subpopulations from the anti-hapten polyclonal antibody repertoire. This served as a tool for the control of assay specificity. Previously, it was demonstrated that this approach is suitable for changing the immunodetection of individual analytes to group-specific recognition, using the same antibody. This was exemplified by 16-membered macrolides [37], amphenicols [18], and fluorophenyl-containing fluoroquinolones [10]. In the present work, using the glycopeptide antibiotics as model substances, we investigated the role of the immunizing hapten exposition on the repertoire of generated antibodies and the efficiency of the antigen-mediated correction of assay specificity.

#### 3.3.1. Anti-BSA-RSM(pi3)

To study the specificity of anti-BSA-RSM(pi), its interaction with homologous (RSM) and heterologous (TPL) hapten conjugates was examined (Figure 3). Regardless of the various designs of RSM-based coating antigens, the specificity of these homologous ELISA formats remained exclusively RSM-selective (Figure 3A–F, left column). The other representatives (TPL, VCM, and ERM) taken in 100–10,000 ng/mL showed negligible inhibition (B/B_0_ > 85%). Thus, the cross-interaction of these structural analogues did not exceed 0.3%. The coupling procedure of the conjugates contributed to the sensitivity changes, which varied in the range of 4.2–35 ng/mL.

Critical change in assay specificity occurred when hapten in coating antigens was substituted with heterologous, TPL (Figure 3A–F, right column). Presentation of TPL molecules on the carrier allowed the selective binding of anti-RSM to structures common for RSM and TPL and even other representatives from this family of substances. Because of the various designs of TPL-conjugates, more optimal TPL orientation on the carrier, for broad specificity, could be found. However, two variants were found to be unsuitable. The spatial image of hapten in Gel(pi)-TPL was alien to such an extent that it could not bind to anti-BSA-RSM(pi3). Interaction with Gel-TPL(edc) was strong; however, it could not be inhibited by a free analyte.

The other four TPL-based ELISA variants demonstrated the different extent of group recognition of related glycopeptides. The cross-reactivity of TPL increased up to 37% (Figure 3F) and 77% (Figure 3A) and, in some cases, became even better than RSM detection (Figure 3D,E). The similar cross-reactivity profile in Figure 3D,E may be evidence of similar TPL orientation in Gel-TPL(ae) and Gel-TPL(f). Therefore, the resorcyl that is the nearest to the carboxylic group was the most likely site of Mannich condensation. The assay based on immobilized Gel-TPL(pi) was characterized by the most pronounced group recognition of glycopeptides. The cross-reactivity of RSM, VCM, TPL, and ERM ranged between 37% and 100%. The coupling of TPL with the carrier, through periodate-oxidized carbohydrates, allowed it to mask the distinctive glycolipid fragment and to expose TPL-epitopes common for glycopeptide antibiotics from both subgroups. A wide working range of assay 10-10,000 ng/mL was adequate for measurement of antibiotics within its therapeutic window of plasma concentrations 10–30 μg/mL (for TPL) [38] and other more active glycopeptides.

Thus, the conjugation of RSM through multiple sites (carbohydrate fragments) contributed to its variable orientation in the immunogen and generation of antibodies with expanded multi-epitope specificity. The subsequent affinity fractionation using differently designed antigens appeared to be very effective.

A similar effect was reported concerning anti-ERM antibodies generated against ERM conjugated with glucose oxidase, using the glutaraldehyde method [16]. Several available amines in the ERM molecule (Figure 1) were likely involved in the conjugation. The resultant multiple orientation of the hapten in the immunogen induced an immune response with expanded specificity towards ERM epitopes. The specificity of immunorecognition could be converted from ERM-selective to ERM/VCM-group depending on the type of the coating hapten. Thus, the multi-presentation of ERM can contribute to effective antigen-mediated changes in the specificity of the assay.

The limit of RSM detection for selective assay variants based on RSM-coated conjugates was an average of 1 ng/mL. The limit of detection for representatives RSM, TPL, ERM, and VCM in group-recognition ELISA was one order of magnitude higher (about 10 ng/mL). Nevertheless, this group-specific assay appeared to be much more sensitive than the receptor-based fluorescence polarization assay with a limit of detection of 0.25–2 µM (~0.5–4 µg/mL) for VCM, TPL, and telavancin and similar solid-phase enzyme-receptor assay [39,40].

#### 3.3.2. Anti-BSA-TPL(ae)

The case of an unambiguous hapten-exposed immunogen will be considered here, exemplified with the cross-reactivity of anti-TPL. Among a variety of coating antigens, only homologous hapten-derived conjugates were capable of binding with Anti-BSA-TPL(ae). The corresponding variants of ELISA were examined and compared for specificity. The binding of antibody with Gel-TPL(ae) and Gel-TPL(edc), the conjugation of which was identical to that of the immunogen, could not be effectively inhibited by free hapten. Homologous assay formats are known to be relatively insensitive in comparison with heterologous ones [41]. The carbodiimide-mediated coupling and active ester methods formed zero-length bonds between hapten and protein (Table 1). Therefore, the protein amino acids adjacent to coupled hapten in conjugates may be involved in the immune response and cause strong binding to haptenized proteins. This case is a manifestation of the too-weak inhibitory activity of free TPL caused by a fairly strong antibody interaction with homologous conjugates. The similar effect of Gel-TPL(ae) and Gel-TPL(edc) is an additional confirmation of their identity.

All four developed ELISAs were based on the TPL-coating antigens of heterologous conjugation (Figure 4). Although various conjugation methods could modify the TPL orientation on the protein, the homologous hapten-based antigens were not effective in changing the specificity of the assay. Their influence on the sensitivity of ELISAs was insignificant, making only two-fold changes (IC_50_ = 3.5–7.4 ng/mL). The comparable influence of hapten conjugation in RSM systems was more pronounced (IC_50_ = 4.2–35 ng/mL). The identified differences may be related to the repertoire of the specificity of the antibodies produced and, therefore, to the presentation of the immunizing hapten. The more unique the hapten presentation, the more uniform the specificity of antibodies, and the more constant their properties are in changing conditions, and vice versa.

The high sensitivity and wide dynamic range 0.2–100 ng/mL (IC_20_-IC_80_) of TPL-immunoassays allowed testing of minimal volumes of biofluids and high degree of dilution, providing the complete elimination of a possible matrix influence.

Cross-reactivity for related glycopeptides VCM, ERM, and RSM was negligible in all the considered tests, which demonstrated selectivity towards TPL. No influence of coating antigen design on anti-TPL binding selectivity was discovered, indicating uniform antibody specificity. Thus, unlike the versatile presentation of RSM in the immunogen, an epitope-specific repertoire of anti-TPL was very limited, as a result of the strict orientation of immunizing hapten.

## 4. Conclusions

The present paper is devoted to the study of the relationship between the spatial presentation of immunizing hapten and the immune response repertoire to select generic recognition system. Glycopeptide antibiotics as model haptens/analytes were presented in immunogens in unique and multiple orientations by means of corresponding conjugation techniques. Anti-TPL polyclonal antibodies generated against the sole-site conjugate BSA-TPL(ae) demonstrated uniform specificity, providing selectivity towards TPL. They were not capable of recognizing the related glycopeptides, despite TPL-coated antigen design. The sensitivity of 4 ELISA variants for TPL varied within 3.5–7.4 ng/mL, and the dynamic range was 0.2–100 ng/mL. RSM was conjugated to BSA through several glycoside sites to be variously oriented on the immunogen. This immunogen evoked a wide repertoire of antibody responses, which can be divided into fractions with different specificities. The first described anti-RSM antibodies allowed selective RSM determination in homologous-hapten ELISA formats, with IC_50_ values in the range 4.2–35 ng/mL. A heterologous coating conjugate, Gel-TPL(pi), was found as a specific binder support for the fraction of anti-RSM antibodies that were capable of recognizing a number of glycopeptides. The selected reagents were the base for group-specific ELISA for detection of RSM, TPL, ERM, and VCM with cross-reactivity of 37–100% and a 10–10,000 ng/mL dynamic range. Thus, the mixed orientation of immunizing hapten, unlike the unique spatial orientation, helps one to expand the repertoire of the immune response and opportunities for the selection of the required fine-specificity agent.

## Figures and Tables

**Figure 1 biosensors-09-00052-f001:**
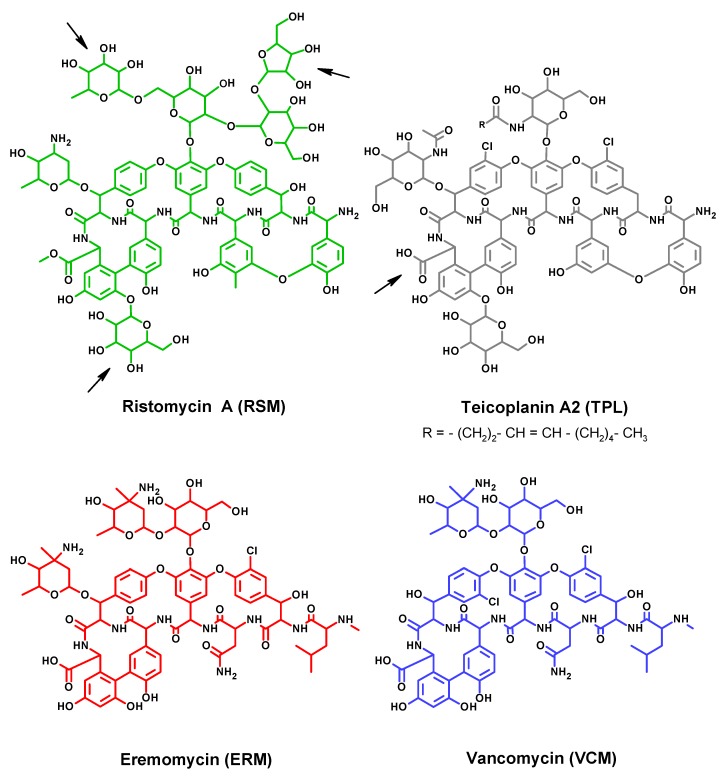
Structures of glycopeptide antibiotics. The sites of possible coupling between glycopeptides and carriers in immunogens are indicated by arrows.

**Figure 2 biosensors-09-00052-f002:**
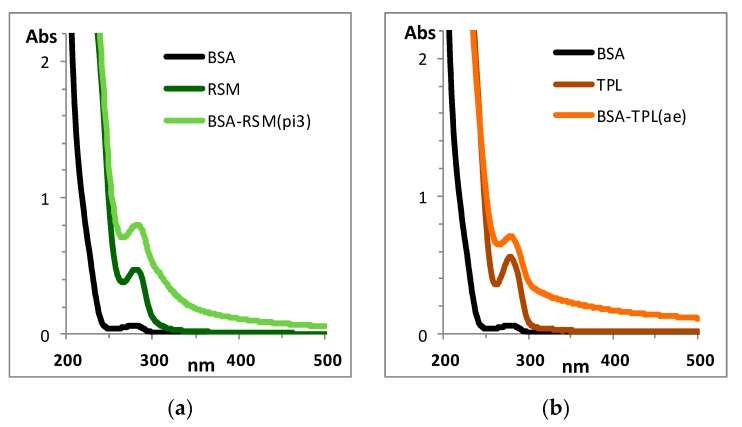
UV-spectra of protein carrier bovine serum albumin (BSA), immunogens, haptens RSM (**a**) and TPL (**b**). Coupling method is indicated in brackets, pi3—periodate (3-fold molar excess) oxidation, ae—active ester method. Concentration of all reagents was 0.1 mg/mL.

**Figure 3 biosensors-09-00052-f003:**
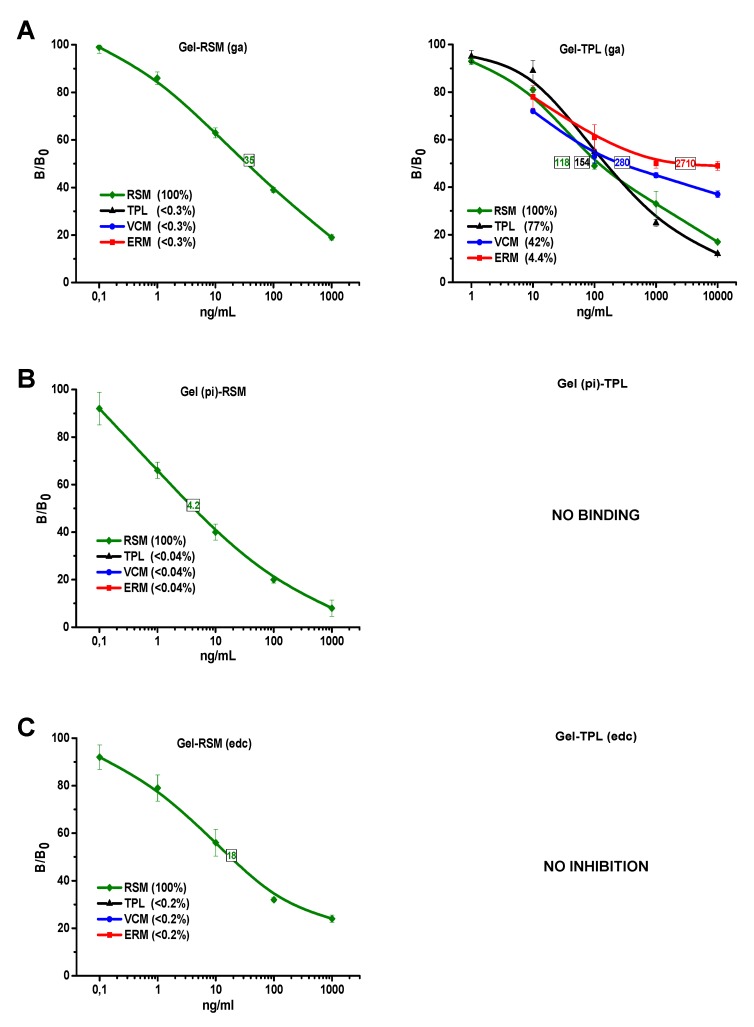
(**A–C**) The influence of coating antigen design (type of hapten and conjugation method) on the recognition spectrum of glycopeptides in immunoassay. The standard curves for indirect competitive enzyme-linked immunosorbent assay (ELISA) variants were derived from interaction between antibodies to BSA-RSM(pi-3) and immobilized conjugates based on ristomycin (RSM) (**left column**) and teicoplanin (TPL) (**right column**). Synthesis of conjugated protein–hapten antigens was carried out using the glutaraldehyde (ga) method (**A**), and reductive amination of the sodium periodate (pi)-oxidized carbohydrates (**B**), using carbodiimide condensation (edc) (**C**). (**D–F**) The influence of coating antigen design (type of hapten and conjugation method) on the recognition spectrum of glycopeptides in immunoassay. The standard curves for indirect competitive ELISA variants were derived from interaction between antibodies to BSA-RSM(pi-3) and immobilized conjugates based on RSM (**left column**) and TPL (**right column**). Synthesis of conjugated protein–hapten antigens was carried out using the active ester (ae) methods (**D**), in Mannich reaction with formaldehyde (f) (**E**) and periodate (pi) oxidation of haptens’ carbohydrate fragments (**F**). The values of IC_50_ (ng/mL) and cross reactivity (%) are given in frames and brackets, respectively. Each symbol indicates the average value (*n* = 3), and the error is represented by SD.

**Figure 4 biosensors-09-00052-f004:**
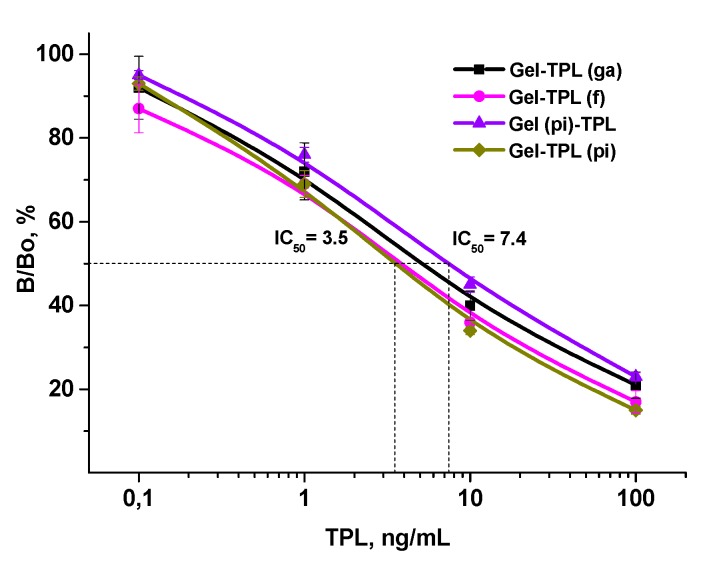
The influence of coating conjugate design (coupling procedure) on teicoplanin (TPL) assay characteristics. The measurements were made in triplicate.

**Table 1 biosensors-09-00052-t001:** Coating conjugates employed in the present study.

Hapten Functional Groups	Coating Conjugates Prepared Using Various Coupling Methods on The Basis of Glycopeptides		Type of Bond Formed Between Protein (pr) and Glycopeptide (gp)
	RSM	TPL	
Amine	Gel-RSM(ga)	Gel-TPL(ga)	[Pr-NH]-(CH_2_)_5_-[NH-Gp]
	Gel(pi)-RSM	Gel(pi)-TPL	[Pr=CH]-[NH-Gp]
Carboxyl		Gel-TPL(edc)	[Pr-NH]-[CO-Gp]
		Gel-TPL(ae)	[Pr-NH]-[CO-Gp]
Phenolic, Resorcylic	Gel-RSM(f)	Gel-TPL(f)	[Pr-NH]-CH_2_-[CH=Gp]
	Gel-RSM(edc) *		
	Gel-RSM(ae) *		
Carbohydrate hydroxyls	Gel-RSM(pi-1)	Gel-TPL(pi-1)	[Pr-NH]-[CH=Gp]
	Gel-RSM(pi-3)	Gel-TPL(pi-3)	

* Phenol is one of the most likely sites of carbodiimide activation when there is no available carboxyl [35].

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
