# Peer review of "Specific and Generic Immunorecognition of Glycopeptide Antibiotics Promoted by Unique and Multiple Orientations of Hapten"

_biosensors, 2019, doi:10.3390/bios9020052_

Round 1
Reviewer 1 Report
In the present manuscript, the authors study the variability in polyclonal immune response obtained after immunization with haptens with defined or variable orientation after coupling to the carrier. They demonstrate diversity in the polyclonal sera obtained following immunization with different variants of haptens and the different level of their cross reactivity to the selected group of immunogens. The study subject is very interesting, but my main comment would be that the authors should reveal substantially more data, such as evaluation of the immune response after the immunization and determination of sensitivity of the ELISA experiments. Further, there are no replicates of measurements shown. The manuscript is carefully written (there are very few typos), but sentence construction is so unusual that the subject matter becomes difficult to understand, and I would politely recommend a read-through by a colleague closer to English language before resubmission. The discussion section is not very long, which could be improved because the finding that different orientational forms of immunogens evoke variable responses is indeed useful and interesting: what would be the effect of alternating the antigen conjugates during the immunization regimen? Additionally, please find enclosed a list of remarks that I hope you would find helpful:
Line 48: maturation methods instead of maturing methods
Line 75: please define the abbreviation D-Ala-D-Ala
Line 102: please specify the conditions of dialysis for all compounds, such as volume of buffer, number of changes and temperature.
Line 145: …against 3 exchanges of 5 L PBS.
Line 148: Liter is sometimes L and sometimes l, please correct throughout the text
Line 167-168:…as described before instead of “ordinary procedure”
Line 180: occurring instead of occurred?
Line 186: This title is too general, maybe “spectroscopic characterization of …”
Line 191: these abbreviations were or should be defined previously in the text
Line 256: RSM?
Line 290: strong interaction instead of high-intensive
Line 296: is the closest most similar?
Line 298-299: This sentence is difficult to understand: did you mean that the results of crossreactivity test was changed when using GEL-TPL-pi for conjugation with the immunogen? Changed from what? Please specify.
Line 299-300: “This assay was characterized”… This sentence is not easy to understand, please reword and explain what the most pronounced properties are.
Line 312: could allow multiple orientation
Line 318 and 326: outcome instead of characteristics
Line 322 and 330: IC50, 50 in subscript
Line 333: the abbreviation LOD is not explained
Figure 3A-F: My interpretation of the figures is that ELISA values for control antigens are given for 2 measurement points (concentration points) only, I would recommend to remove the interpolation (fit) line for those data
Lines 350-352: please reword, what are the micro-volumes of test samples?
Line 353: I think sera are cross-reactive, not the antigens
Line 369: in the range of
Line 370: …has been found to be able to immobilize a fraction of antibodies that…are able to specifically detect all of the related test haptens?
Author Response
Response to the reviewers
The authors are grateful to the editor and the reviewers for taking time, attentive reading, and evaluation of our manuscript. The major revision has been made and all the comments have been considered. Corresponding responses have been attached after referee comments and marked in BLUE. The corrections made in the manuscript are marked in RED. We hope that revised manuscript is acceptable now, and look forward to it being published in Biosensors.
REVIEWER REPORT(S):
Referee: 1
Comments and Suggestions for Authors
In the present manuscript, the authors study the variability in polyclonal immune response obtained after immunization with haptens with defined or variable orientation after coupling to the carrier. They demonstrate diversity in the polyclonal sera obtained following immunization with different variants of haptens and the different level of their cross reactivity to the selected group of immunogens. The study subject is very interesting, but my main comment would be that the authors should reveal substantially more data, such as evaluation of the immune response after the immunization and determination of sensitivity of the ELISA experiments. Further, there are no replicates of measurements shown. The manuscript is carefully written (there are very few typos), but sentence construction is so unusual that the subject matter becomes difficult to understand, and I would politely recommend a read-through by a colleague closer to English language before resubmission. The discussion section is not very long, which could be improved because the finding that different orientational forms of immunogens evoke variable responses is indeed useful and interesting: what would be the effect of alternating the antigen conjugates during the immunization regimen? Additionally, please find enclosed a list of remarks that I hope you would find helpful:
Thank you for your interest in our research and the desire to improve the quality of the article. We are grateful for your comments which were valuable and helpful. The manuscript was edited by the highly qualified native English speaking editors. Their recommendations were almost fully accepted by authors, so neither the research content nor the authors' intentions were altered. Immunization procedure, estimation of immune response, selection of the most sensitive antisera are detailed in (Lines 274-280). Criterion of LOD and dynamic range of assay are supplemented in Lines 197-199.
Discussion of results are enlarged. The additional fragments with comparison, explanations and conclusions are highlighted with RED.
We hope that the revised version of manuscript is acceptable now for publishing in Biosensors.
Line 48: maturation methods instead of maturing methods
Thank you for correction. This change is made.
Line 75: please define the abbreviation D-Ala-D-Ala
The abbreviation is inserted in the place of the first mention
Line 102: please specify the conditions of dialysis for all compounds, such as volume of buffer, number of changes and temperature.
The dialysis was conducted uniformly for all prepared conjugates. So, the common conditions (duration, temperature, volumes of buffer) are indicated on Lines 148-150, after the description of conjugate synthesis.
Line 145: …against 3 exchanges of 5 L PBS.
Line 148: Liter is sometimes L and sometimes l, please correct throughout the text
The recommendations are taken into account. The whole manuscript is checked and corrections of liter, milliliter and microliter are made.
Line 167-168:…as described before instead of “ordinary procedure”
Line 180: occurring instead of occurred?
The changes are done.
Line 186: This title is too general, maybe “spectroscopic characterization of …”
Thank you for your opinion. Maybe “Examination” sounds too general. However, the coating conjugates were also examined on their binding activity with antibody in ELISA. Formation of these conjugates was firstly confirmed immunochemically (Lines 218-219) and additionally using spectroscopy. So, the title is modified to “Synthesis and characterization of conjugated antigens”
Line 191: these abbreviations were or should be defined previously in the text.
The abbreviations are firstly mentioned in ‘Chemicals’ section. The excessive definitions were excepted from section ‘Synthesis and characterization of conjugated antigens’.
Line 256: RSM?
This misprint is revised.
Line 290: strong interaction instead of high-intensive.
Thank you for your corrections
Line 296: is the closest most similar?
Line 298-299: This sentence is difficult to understand: did you mean that the results of crossreactivity test was changed when using GEL-TPL-pi for conjugation with the immunogen? Changed from what? Please specify.
Line 299-300: “This assay was characterized”… This sentence is not easy to understand, please reword and explain what the most pronounced properties are.
The above sentences are paraphrased. Lines 297-302.
Line 312: could allow multiple orientation
The sentences are paraphrased
Line 318 and 326: outcome instead of characteristics
The revised caption is ‘The influence of coating antigen design (type of hapten and conjugation method) on recognition spectrum of glycopeptides in immunoassay.’
Line 322 and 330: IC50, 50 in subscript
Done
Line 333: the abbreviation LOD is not explained.
The explanation, abbreviation and corresponding citation are inserted in ELISA procedure section, Lines 188-190.
Figure 3A-F: My interpretation of the figures is that ELISA values for control antigens are given for 2 measurement points (concentration points) only, I would recommend to remove the interpolation (fit) line for those data.
We demonstrated inhibition activity of TPL, VCM and ERM which were more than 85% for 100-10000 ng/mL range of concentrations for better visual comparison between two RSM-based and TPL-based ELISA formats. The corresponding explanation is in Lines 286-288
Lines 350-352: please reword, what are the micro-volumes of test samples?
The phrase is reworded
Line 353: I think sera are cross-reactive, not the antigens.
In our opinion, interaction is a mutual process. One antibody can react with several antigens as well as several antigens can react with the same antibody. A lot of confirming examples can be found in the literature, i.e ‘Current understanding of cross‐reactivity of food allergens and pollen’ https://doi.org/10.1111/j.1749-6632.2002.tb04132.x ; ‘Absence of cross-reactivity between sulfonamide antibiotics and sulfonamide nonantibiotics’ DOI: 10.1056/NEJMoa022963; ‘The cross-reactivity and immunology of β-lactam antibiotics’ DOI: https://doi.org/10.2165/00002018-199410040-00006; ‘Validation and Cross-Reactivity Data for Fentanyl Analogs With the Immunalysis Fentanyl ELISA’ https://doi.org/10.1093/jat/bky060;
Line 369: in the range of
Done
Line 370: …has been found to be able to immobilize a fraction of antibodies that…are able to specifically detect all of the related test haptens?
The phrase is reworded

Reviewer 2 Report
The question of group-selective antibodies should be discussed in a broader sense. Some more general citations are required. Particularly, the polyclonality of the sera makes the discussion relatively complex.
"Phenyl is one of the most likely sites of carbodiimide activation when there is no available carboxyl." I never heard from this reaction. Even Mannich reaction will require activated carbon atoms. Maybe the authors meant "phenol" instead of "phenyl"? Perhaps some reaction schemes should be shown.
In Fig. 3A-C the non-inhibiting curves are shown in an uncommon form. the black, blue and red curves should not be shown at all, if no real data are available or should be shown. The other alternative would be to show the whole graph from 0.1 to 1000 ng/ml. BTW; µg/L would be the preferable unit throughout.
The authors compared homologous and heterologous combinations. However, the respective concept was not clearly described, discussed or cited.
In respect to the different orientations of the haptens, I cannot see a big difference.
"Antibody binding to Gel-TPL(ae) and Gel-TPL(edc), conjugation of which was identical to that in immunogen could not be inhibited by free hapten." Do you have any explanation for this?
In addition, the effect of repeated immunizations would be interesting to show. Some researchers claim that even with a heterogenous antigen, with repeated immunizations, the sera would become more and more narrow in their epitope specificity ("pseudo-monoclonal"). Could this be observed here too?
I think that the authors did not clearly differentiate between a true cross-reactivity (which also occurs in monoclonals) and the shared reactivity which is only present in polyclonal antibodies and could be directed against completely unrelated epitopes.
The conclusions were not very clear, particularly in respect of the title, in which the orientation is a major topic.
Author Response
Response to the reviewers
The authors are grateful to the editor and the reviewers for taking time, attentive reading, and evaluation of our manuscript. The major revision has been made and all the comments have been considered. Corresponding responses have been attached after referee comments and marked in BLUE. The corrections made in the manuscript are marked in RED. We hope that revised manuscript is acceptable now, and look forward to it being published in Biosensors.
Referee: 2
Comments and Suggestions for Authors
The question of group-selective antibodies should be discussed in a broader sense. Some more general citations are required. Particularly, the polyclonality of the sera makes the discussion relatively complex.
Thank you for your careful consideration of the manuscript, comments and good advices.
A number of approaches to develop antibody for broad recognition are considered in the paper.
They are based on targeted immunization, screening/artificial maturation and combination (Lines 47-52).
An additional point concerning polyclonal antibodies and strategy of their affinity fractionation for development of group-selective assays was added in introduction (Lines 53-58). Confirming citations are included [15-18]. The main principle of affinity fractionation approach is considered (in Lines 283-286), is compared with the previous results (in Lines 343-350) and is discussed using the models RSM vs TPL (in Lines 364-383).
"Phenyl is one of the most likely sites of carbodiimide activation when there is no available carboxyl." I never heard from this reaction. Even Mannich reaction will require activated carbon atoms. Maybe the authors meant "phenol" instead of "phenyl"? Perhaps some reaction schemes should be shown. Thank you for your remark. Of course we mean fenolic instead of fenyl group. This blunder was avoided. In our experiments we use classical bioconjugate techniques and well-known reactions. The results of these conjugations are described in detail in 3.1 section. The corresponding citations are included as well. The exception is the case with carbodiimide activation of RSM that have no -COOH, a typical target for edc. This case is discussed separately in Lines 212-217, and possible but not exact site of interaction was proposed based on literature data [30]. The possible hapten sites taking part in immunogen preparations are shown in Fig 1. The expected constructs based on hapten functional groups and the carrier are presented in Table 1. Thus, on the one hand, there is no need to show well-known reactions from the handbook [31]. From other side, we cannot show reaction schemes based on our suspicions.
In Fig. 3A-C the non-inhibiting curves are shown in an uncommon form. the black, blue and red curves should not be shown at all, if no real data are available or should be shown. The other alternative would be to show the whole graph from 0.1 to 1000 ng/ml. BTW; µg/L would be the preferable unit throughout.
The real data for TPL, VCM and ERM were between 100% and 85% for 100-10000 ng/mL range of concentrations. Our initial trial to present real data resulted in indistinguishable overlapping curves.
So, for better visual comparison of inhibition activity between RSM-based and TPL-based ELISAs formats we demonstrated the simulated non-inhibiting curves. According to your recommendations they are not shown now. The explanation of their absence is added in Lines 286-288.
The authors compared homologous and heterologous combinations. However, the respective concept was not clearly described, discussed or cited. The first mention of hapten heterology effect is introduced in Lines 51-59 with citation of previous work on glycopeptides. Then, the main principle of in situ selection of antibody to common epitopes using heterology hapten approach is presented at the end of introduction (Lines 87-96).
In respect to the different orientations of the haptens, I cannot see a big difference.
Different orientations of hapten in immunogen help us to present hapten for immune response more variously and provide the expanded repertoire of anti-hapten specificities. Wide repertoire presents more possibilities for selection a required candidate.
One big difference (concerning sole and multiple orientation haptens in immunogen) is the difference between TPL- and RSM-specificity (narrow and wide).
Another big difference (concerning TPL orientation on coating antigens) is the difference between cross-reactivity profiles for assay systems in right column recognition.
"Antibody binding to Gel-TPL(ae) and Gel-TPL(edc), conjugation of which was identical to that in immunogen could not be inhibited by free hapten." Do you have any explanation for this?
It is very often situation when the homologous antigen which is identical to immunogen shows too strong binding to antibody to be inhibited by free hapten. So it is not suitable for competitive assay. That is why the well-known effect of heterology of hapten, conjugation method, linker or spacer is often used to weak such binding and to improve an inhibitory activity of analyte. Unfortunately, we know few analogical cases from the literature, due to rejection of such antibodies during selection procedure.
In addition, the effect of repeated immunizations would be interesting to show. Some researchers claim that even with a heterogenous antigen, with repeated immunizations, the sera would become more and more narrow in their epitope specificity ("pseudo-monoclonal"). Could this be observed here too?
We studied the activity and sensitivity in dynamics of immunization only in homologous-hapten-based ELISA formats to choose the most sensitive variant. So, we cannot present here data on cross-reactivity.
In our several attempts with other haptens where we studied this question we reveal no significant changes in cross-reactivity profile, only changing in sensitivity. We think that the mentioned narrowing of specificity can be observed towards immunodominant and more immunogenic antigens in the mixture.
I think that the authors did not clearly differentiate between a true cross-reactivity (which also occurs in monoclonals) and the shared reactivity which is only present in polyclonal antibodies and could be directed against completely unrelated epitopes.
We judge about cross-reactivity only for a certain system of interaction (Ab-Ag). McAb or PcAb.
When Ag is changed, CR may be changed too, more prominent for PcAb than for McAb.
For PcAb the effect of heterologous antigen is double. CR change is caused by fractionation of PcAb. As a result, the selected oligo- or even “pseudo-monoclonal" antibodies show a corresponding CR profile.
The conclusions were not very clear, particularly in respect of the title, in which the orientation is a major topic.
Conclusion is revised for more focus on relationship ‘hapten orientation - assay specificity’
The Authors appreciate greatly the Reviewers for all corrections and comments which are valuable and helpful to improve the quality of the manuscript.
Round 2
Reviewer 1 Report
The manuscript has improved with the revision and teh authors have responded to reviewer's remark. Please label the Y-axis in Figure 2 and replace decimal commas for points (I am sorry I did not notice that in the first version).